# A Real-World Longitudinal Study in Non-Functioning Pituitary Incidentalomas: A PRECES Micro-Adenomas Sub-Analysis

**DOI:** 10.3390/diseases12100240

**Published:** 2024-10-02

**Authors:** Mihai Costachescu, Claudiu Nistor, Ana Valea, Oana-Claudia Sima, Adrian Ciuche, Mihaela Stanciu, Mara Carsote, Mihai-Lucian Ciobica

**Affiliations:** 1Department of Radiology and Medical Imaging, Fundeni Clinical Institute, 022328 Bucharest, Romania; mihaicostachescu@gmail.com; 2PhD Doctoral School, “Carol Davila” University of Medicine and Pharmacy, 010825 Bucharest, Romania; 3Thoracic Surgery Department, “Dr. Carol Davila” Central Emergency University Military Hospital, 010825 Bucharest, Romania; claudiu.nistor@umfcd.ro (C.N.); adrian.ciuche@umfcd.ro (A.C.); 4Department 4—Cardio-Thoracic Pathology, Thoracic Surgery II Discipline, “Carol Davila” University of Medicine and Pharmacy, 0505474 Bucharest, Romania; 5Department of Endocrinology, “Iuliu Hatieganu” University of Medicine and Pharmacy, 400012 Cluj-Napoca, Romania; 6Department of Endocrinology, County Emergency Clinical Hospital, 400347 Cluj-Napoca, Romania; 7Department of Endocrinology, “Lucian Blaga” University of Sibiu, Victoriei Blvd., 550024 Sibiu, Romania; mihaela.stanciu@ulbsibiu.ro; 8Department of Endocrinology, Clinical County Emergency Hospital, 550245 Sibiu, Romania; 9Department of Endocrinology, “Carol Davila” University of Medicine and Pharmacy, 050474 Bucharest, Romania; carsote_m@hotmail.com; 10Department of Clinical Endocrinology V, C.I. Parhon National Institute of Endocrinology, 011863 Bucharest, Romania; 11Department of Internal Medicine and Gastroenterology, “Carol Davila” University of Medicine and Pharmacy, 020021 Bucharest, Romania; lucian.ciobica@umfcd.ro; 12Department of Internal Medicine I and Rheumatology, “Dr. Carol Davila” Central Military University Emergency Hospital, 010825 Bucharest, Romania

**Keywords:** tumour, computed tomography, pituitary, hormone, endocrine, follow up, incidentaloma

## Abstract

**Background.** Incidentalomas have an increasing incidence all over the world due to a larger access to imaging assessments, and endocrine incidentalomas make no exception in this matter, including pituitary incidentalomas (PIs). **Objective.** Our objective was to analyse the dynamic changes amid a second computed tomography (CT) scan after adult patients were initially confirmed with a PI (non-functioning micro-adenoma). **Methods.** This was a multi-centric, longitudinal, retrospective study in adults (aged between 20 and 70 y) amid real-world data collection. We excluded patients who experienced baseline pituitary hormonal excess or deficiency or those with tumours larger than 1 cm. **Results.** A total of 117 adults were included (94.02% females) with a mean age of 43.86 ± 11.99 years, followed between 6 and 156 months with a median (M) of 40 months (Q1 Q3: 13.50, 72.00). At the time of PI diagnosis, the transverse diameter had a mean value of 0.53 ± 0.16 cm, the longitudinal mean diameter was 0.41 ± 0.13 cm, and the largest diameter was 0.55 ± 0.16 cm. No PI became functioning during follow-up, neither associated hypopituitarism nor increased >1 cm diameter. A total of 46/117 (39.32%) patients had a larger diameter during follow-up (increase group = IG) versus a non-increase group (non-IG; N = 71, 60.68%) that included the subjects with stationary or decreased diameters. IG had lower initial transverse, longitudinal, and largest diameter versus non-IG: 0.45 ± 0.12 versus 0.57 ± 0.17 (*p* < 0.0001), 0.36 ± 0.11 versus 0.43 ± 0.13 (*p* = 0.004), respectively, 0.46 ± 0.12 versus 0.6 ± 0.16 (*p* < 0.0001). IG versus non-IG had a larger period of surveillance: M (Q1, Q3) of 48 (24, 84) versus 32.5 (12, 72) months (*p* = 0.045) and showed similar age, pituitary hormone profile, and tumour lateralisation at baseline and displayed a median diameter change of +0.14 cm versus −0.03 cm (*p* < 0.0001). **To conclude**, a rather high percent of patients might experience PI diameter increase during a longer period of follow-up, including those with a smaller initial size, while the age at diagnosis does not predict the tumour growth. This might help practitioners with further long-term surveillance protocols.

## 1. Introduction

Incidentalomas have an increasing incidence all over the world due to a larger access to imaging assessments, and endocrine incidentalomas make no exception in this matter [1,2,3]. Pituitary incidentalomas (PIs) stand for the most common incidentalomas of the endocrine glands, other than thyroid incidentalomas (meaning thyroid nodules), a term that is not actually very often used in daily practice [4,5,6]. The clue of applying the term “incidentalomas” relates to the accidental radiological detection amid performing an imaging scan for unrelated purposes to the actual endocrine tumour, for example, in patients with PIs, headaches, neurological conditions, trauma, screening protocols for malignancies, etc. [7,8,9].

PIs affect one out of ten people; they are mostly non-functioning micro-adenomas, meaning the largest diameters are less than 1 cm [10,11,12]. The most frequent hormonal excess (if any) is of prolactin, and then usually the tumour will be named as such (prolactinoma) rather than PI during further monitoring [13,14,15]. The rate of significant tumour growth is 10% for micro-PIs and up to 25–30% for macro-PIs, but many heterogeneous results have been published so far with regard to the spontaneous tumour behaviour [16,17,18].

Initial assessment includes, in addition to the scan that provided the radiological identification of the tumour, such as computed tomography (CT) or magnetic resonance imaging, the hormonal panel with respect to the pituitary hormones as well as an eye exam (more important in macro-adenomas) [19,20,21]. The timing of re-assessment is still an open matter, including the protocols of serial re-scans across a life span that varies between centres [22,23,24]. The management is conservative for micro-PIs and for most of the macro-PIs (but not all) [25,26,27].

Our objective was to analyse the dynamic changes amid a second CT scan after adult patients were initially confirmed with a PI (non-functioning micro-adenoma) based on a CT exam and an endocrine evaluation.

Our hypothesis was that during more than 3 years of mean follow-up duration of imagistic surveillance, some patients might experience micro-adenoma increase, including in smaller tumours, but no significant clinical impact is expected (in terms of remaining non-functioning, not becoming neurosurgery candidates due to massive PI increase, and becoming larger than 1 cm or associating a pituitary apoplexy).

## 2. Materials and Methods

### 2.1. Study Design

This was a multi-centric, multi-disciplinary, longitudinal, retrospective study in adults who were diagnosed with a PI (non-functioning hypohyseal micro-adenoma) amid real-world data collection.

### 2.2. Studied Population

We included asymptomatic patients (aged between 21 and 70 years) who were confirmed with a non-functioning pituitary micro-adenoma that was detected respecting the scenario of a PI (cross-sectional analysis), and then the individuals had a second CT scan that was performed between June 2019 and July 2024 (longitudinal analysis). Inclusion criteria were age of 18 years and older, the patients signed the informed consent according to each hospital rules (during hospitalisation for imagery and endocrine assessments), the PI was accidentally detected while the subject underwent a CT scan for unrelated purposes (e.g., headache, sinusitis, different ophthalmic or neurologic ailments, etc.), and data being available with concern to the second CT scan and endocrine evaluation. Exclusion criteria were functional pituitary tumours, active endocrine tumours of any location, suspicion of a pituitary malignancy (primary or secondary), hypothalamic tumours, lack of complete hormonal assessment to prove the non-functioning pattern of the PI, tumour size at least or larger than 1 cm diameter at initial CT scan, prior neurosurgery of any type, hypopituitarism at initial evaluation, double or triple PIs, cystic pituitary lesions or tumours that were suggestive for other histological (non-adenoma) types, and active cancers of any primary origin.

### 2.3. Study Protocol

Patients who were diagnosed as having a PI (non-functioning micro-adenoma) based on the CT scan and the hormonal assessment underwent a second re-assessment following at least 6 to 12 months since the initial diagnosis, depending on their medical and surgical background. This was based on an individual decision for each case according to their current physician, who followed the patient for the second evaluation as well.

#### 2.3.1. Imaging Scans

The imagery investigation was based on intravenous contrast CT scans at the first and second times (the follow-up period was between 6 and 156 months). CT-based parameters were transverse and longitudinal diameters (and the term “largest” diameter was applied as provided by these two diameters) and the location of the tumour (left, right, or median). A second check-up CT analysis was further processed (by Dr. MC) after imaging data were registered in each centre in order to confirm the PIs size and achieve a homogenous interpretation of all the data included in this study.

#### 2.3.2. Hormonal Assays

The endocrine panel included the pituitary hormone assays, namely follicle-stimulating hormone (FSH), luteinising hormone (LH), TSH (thyroid stimulating hormone), growth hormone (GH), adrenocorticotropic hormone (ACTH), and prolactin, as well as the peripheral hormones such as plasma morning cortisol and insulin-like growth factor 1 (IGF1). All these mentioned endocrine assays were performed during the first and second evaluations at the moment of the CT scan. In selected cases, the individual physician of one patient decided to perform a dynamic hormonal test (either of suppression or stimulation), and the subjects who showed any type of an endocrine anomaly (either complete or partial deficiency or excess) at baseline evaluation were not included. If a PI proved to be a functioning tumour during follow-up, this case was excluded from the final analysis (Figure 1).

### 2.4. Statistical Analysis

The data were collected and processed using Excel 16.78 and statistically analysed using SPSS 29.0.2.0. Normality tests (Kolmogorov–Smirnov) and empirical observation were used to assess the distribution type of each continuous variable. Central tendency measures were presented as mean ± standard deviation (SD) or as quartiles (Q1, median/Q2, Q3). The chi-squared test and Fisher’s exact test were applied to explore associations between categorical variables. For comparisons of continuous variables, the Student’s *t*-test was used for those with a normal distribution and the Mann–Whitney U test for those without. Correlations between variables were assessed using Pearson’s correlation coefficient for normally distributed data and Spearman’s correlation coefficient for non-normally distributed data. To compare multiple groups, an analysis of variance test (ANOVA) was utilised for normally distributed data, while the Kruskal–Wallis test was used for non-normal data. Receiver operating characteristic (ROC) curves were plotted to analyse numerical variables related to outcomes, and the area under the curve (AUC) was calculated to measure performance. The Youden index was determined to find the optimal cut-off value. A Kaplan–Meier survival analysis was used to create growth-free survival curves. A statistical significance level of under 0.05 was used (*p* < 0.05).

### 2.5. Ethical Aspects

The subjects signed an informed consent. This study was conducted in accordance with the Declaration of Helsinki. This is a sub-analysis of the PRECES study (parameters of Romanian population with endocrine conditions with or without endocrine surgery: real-world evidence and retrospective study), a multi-centric collaborative in the field of endocrinology and associated domains. Ethical Committees approved the retrospective data collection with regard to prior hospitalised patients who were diagnosed with PIs (702-28.06.2024; 665-31.01.2024; 124-25.06.2024; 6284-08.02.2024; 2058-30/01/2024).

## 3. Results

### 3.1. Cross-Sectional (Baseline) Analysis

A total of 117 patients were evaluated in this study after applying the inclusion and exclusion criteria. Of these, 94.02% were females and 5.98% were males, with a mean age of 43.86 ± 11.99 years for the entire studied population. At the time of PI diagnosis, the transverse diameter at the CT scan had a mean value of 0.53 ± 0.16 cm, the longitudinal mean diameter was 0.41 ± 0.13 cm, and the largest measured diameter was 0.55 ± 0.16 cm. Regarding location/lateralisation analysis, 32.48% of PIs were on the left side, 47.01% were on the right side, and 20.51% were in the median part of the pituitary fossa.

FSH levels had a median of 33.40 mIU/mL (of note, we included the female patients regardless of their menopausal status); LH had a median of 5.66 mIU/mL. The mean ACTH value was 20.57 ± 13.09 pg/mL, and the mean basal plasma morning cortisol level was 12.79 ± 3.87 µg/dL. The median GH was 0.30 ng/mL, the average IGF-1 level was 167.28 ± 42.31 ng/mL, and the average prolactin was 8.24 ± 4.46 ng/mL (Table 1).

### 3.2. Analysis Based on the Patients’ Age Groups

The analysis based on patients’ decades of age showed that most patients were within the decades 31–40 years (23%), 41–50 years (25%), and 51–60 years (28%) (Table 2).

After dividing the studied population into five age groups, the median largest PI diameter showed no statistically significant difference between these groups (ANOVA test, *p* = 0.334) (Figure 2).

Regarding the PIs side amid patients’ age groups, in the 31–40 years age group, there were significantly more PIs located on the left side than expected (+2.0 adjusted residual) and fewer on the right side (−2.5 adjusted residual). In the 41–50 years age group, there were more PIs located in the right part than expected (+2.1 adjusted residual, *p* = 0.045) (Figure 3).

### 3.3. Longitudinal Analysis

The cumulative probability of having a subsequent CT scan had a sharp decrease from 0.93 to 0.75 at 12 months following the baseline CT scan. After 12 months, the cumulative probability of having another CT scan had a more gradual decrease, extending to the endpoint at 157 months (Figure 4).

When comparing the baseline CT’s largest diameter with the diameter change between the baseline and the 2nd CT, a statistically significant negative correlation was found (r = −0.575, *p* = 0.000) (Figure 5).

#### 3.3.1. Analysis between the Baseline (First) CT Scan and the Second (2nd) CT Scan

In terms of the tumour size change, 39.32% (N = 46/117) of the entire group had an increase in the largest PI diameter (increase group) compared with 60.68% of the participants from the initial cohort that had PIs with an unchanged or decreased largest diameter (non-increase group). The largest PI diameter was statistically significantly smaller at the baseline versus the 2nd CT scan in the increase group (*p* < 0.005 for each), respectively, larger at the baseline versus the follow-up in the non-increase group (*p* < 0.001 for each) (Table 3).

#### 3.3.2. Analysis between the Increase Group versus the Non-Increase Group

The age was similar between the two groups. The transverse diameter was statistically significantly smaller in the increase group versus the non-increase group at the baseline CT (*p* = 0.000). After the 2nd CT scan, the transverse diameter was larger for the increase group (*p* = 0.000). The longitudinal PI diameter was significantly smaller in the increase group versus the non-increase group at the baseline evaluation (*p* = 0.004), and for the 2nd CT, it was similar between both groups. At the baseline CT, the largest PI diameter was smaller in the increase group compared to the non-increase group (*p* = 0.000); at each patient’s 2nd CT, the largest diameter was larger in the increase group versus the non-increase group (*p* = 0.000). The diameter change between the largest diameter at the initial and 2nd CTs was higher for the increase group versus the non-increase group. The median diameter change was +0.14 cm in the increase group versus −0.03 cm in the non-increase group (*p* = 0.000) (Figure 6).

PIs side distribution was similar between the groups (*p* = 0.088), as were the hormonal panel and the age of the patients. Any of the diameters was statistically significant different between the groups at the baseline and during follow-up (Table 4).

#### 3.3.3. Analysis of Time Window between Assessments

The follow-up period was higher (*p* = 0.045) in the increase versus the non-increase groups with a median of 50 months (interquartile interval between 24 and 84), respectively, of 24 months (interquartile interval between 12 and 72) (Table 5).

#### 3.3.4. Arbitrary Cut-Off and Receiver Operating Characteristic (ROC) Curve to Predict PI Increase

An arbitrary cut-off value of 0.50 cm was chosen to form two risk groups: a group with the largest baseline diameter smaller than 0.50 cm and a group with the largest baseline diameter larger or equal to 0.50 cm. When comparing the groups with the largest diameter, <0.50 cm and ≥0.50 cm, with the increase and non-increase groups, a statistically significant association was found between the <0.50 cm group and the increase group, respectively, between the ≥0.50 cm and the non-increase group (*p* = 0.000) (Table 6).

The odds ratio was estimated, and the <0.50 cm group had a 5.52 (95% C.I., 2.46–12.40) times higher risk of increasing diameter when compared to the ≥0.50 cm group (*p* = 0.000). Using the largest diameter at the baseline CT scan to predict an increase in the PI diameter, the receiver operating characteristic curve (ROC) was plotted using smaller values to indicate stronger predictive evidence for a positive outcome (AUC = 0.757, *p* = 0.000). To identify the most appropriate cut-off diameter, the Youden index was calculated for various potential cut-off points. The optimal cut-off diameter was determined to be 0.545 cm, which provided a sensitivity of 87.00% and a specificity of 59.20%. By comparison, the arbitrary cut-off point of 0.5 cm yielded a lower Youden index, with a smaller sensibility of 65.21% and a higher specificity of 74.64% (Table 7).

As noted in Table 4, the baseline largest CT diameter may be the strongest predictor for PI growth; therefore, using the baseline CT diameter, we plotted the ROC curve: for each possible cut-off point, we generated the sensitivity and 1-specificity chart with a significant area under the curve of 0.757, demonstrating the strength of the baseline largest CT diameter as a predictor. The ROC curve shows the sensitivity and specificity for the PI increase (Figure 7).

## 4. Discussion

The domain of incidentalomas represents a very important topic amid the modern medicine era due to a more facile access to imaging investigations in both the paediatric and adult populations [28,29,30]. When it comes to PIs, more data have been provided by magnetic resonance imaging rather than CT scans due to their non-irradiating profile, but a CT scan may be more accessible in some centres, or the underlying medical/surgical condition may benefit from a CT scan [1,28,29,30]. Of note, we used the term “micro-adenoma”, but the most recent WHO (World Health Organization) classification switched prior “pituitary adenomas” into “pituitary neuroendocrine tumour” (PitNET) in addition to multiple other terminology/clusters of classification changes [31].

In this study, we analysed 117 adults diagnosed with PIs amid an accidental CT scan (mostly a female population) and confirmed the previously mentioned hypothesis. The patients (aged between 20 and 70 years) were followed from 6 to 156 months when a second CT scan and an endocrine re-assessment were performed. This aspect varied with the patient since no distinct protocol of surveillance was applied, as expected in real-life medicine. No PI patient proved to have a functioning adenoma during follow-up, nor experienced hypopituitarism as shown by the clinical evaluation and the blood hormone assays during the second evaluation. However, we mention that dynamic testing for the pituitary function was only selectively carried out based on the clinical panel and not routinely assessed. Moreover, we did not register any tumour increase beyond the diameter of 1 cm (nor pituitary apoplexy), and this might explain why no case of hypopituitarism was identified after the surveillance period.

Other recent studies showed a heterogeneous spectrum of results. For example, the UK Non-functioning Pituitary Adenoma Consortium included 459 subjects who were confirmed with micro-PIs (with a median age of 44 years), and 419/459 of them had at least one second radiological assessment performed, which in this instance was magnetic resonance imaging (follow-up median of 3.5 years). The authors confirmed that 1/419 cases developed pituitary apoplexy; the rate of growth was 7.8% after 3 years, respectively, 14.5% after 5 years of follow-up, with reduction rates of 14.1%, respectively, of 21.3% for the same time frame. The median change diameter was 0.2 cm [1]. We found a median diameter change of +0.14 cm versus −0.03 cm in the increase versus non-increase groups, *p* = 0.000. In the mentioned study, among the PIs that increased their size, almost half of them became macro-adenomas and had >0.5 cm at baseline, while 1.9% of all the patients underwent neurosurgery [1]. As seen in our cohort, the age at baseline was not a predictor of tumour growth. While we did not include patients who initially experienced any type of hypopituitarism, in this cohort, they had almost 10% of the subjects affected by different types of central hormonal insufficiencies (the most frequent being hypogonadism) at baseline and another 0.6% during follow-up (due to macro-adenoma growth) [1].

We found that 46/117 (39.32%) patients had a larger diameter during follow-up (increase group). This group had a statistically significant lower initial transverse, longitudinal, and largest diameter versus the non-increase group (that included patients with stationary and decreased diameter)—*p* < 0.005 for each diameter. However, the increase group had a longer surveillance period, a mean of 58.11 versus 46.28 months in the non-increase group (*p* = 0.045). Similarly, Jung et al. [27] showed in 246 patients diagnosed with asymptomatic non-functioning PIs (enrolled between 2007 and 2023) that 33/245 subjects displayed an increase in PI size over a mean imaging follow-up period of 27.3 months, and 10 out of the 33 individuals became neurosurgery candidates due to compressive effects [27]. Notably, we did not have in our cohort subjects who became surgery candidates based on the tumour growth.

The adequate timing of repeating the imaging scan varies. For instance, a UK survey published in 2023 showed that recently, more than 80% of the endocrine practitioners were more likely to stop the surveillance at or before 36 months in subjects diagnosed with asymptomatic non-functioning micro-adenomas (PIs) [23]. Yet, based on our data, 39% of the patients registered a diameter increase upon a median follow-up of 48 months; thus, discharging from pituitary monitoring might not be applicable in certain individuals.

As limits of this study, we mention the real-world data collection that may bring a level of bias since no specific surveillance protocol was used in each subject. Also, the selective use of the pituitary function dynamic testing implied that some cases with partial hormonal deficiency and asymptomatic presentation might be missed. Our retrospective data collection did not allow observing all the circumstances for which the CT scan was initially performed (such as trauma, infections, headache, etc.). The metabolic profile and sex distribution were generally not found to be predictors of PI growth [1,32,33]. We did not have enough data to analyse these particular aspects. The study cohort was overwhelmingly female (94.02%), which might limit the generalisability of the findings to male patients.

To our knowledge, the major strength of this study is represented by the fact that a higher percent of patients might increase the PI diameter during a longer period of follow-up, including those with a smaller initial size (while the age at diagnosis does not predict the tumour growth), and further larger trials might mitigate this aspect.

## 5. Conclusions

To summarise:This was a longitudinal study in 117 adults (aged between 20 and 70 years) diagnosed with non-functioning micro-PIs followed for a mean period of 50.93 months.No PI became functioning during follow-up, nor associated hypopituitarism or increased beyond the diameter of 1 cm; no case of pituitary apoplexy was found.The analysis based on patients’ decades of age showed that most of them were between the ages of 31 and 60 with similar diameters.A total of 46/117 (39.32%) patients had a larger diameter during follow-up (increase group) versus the non-increase group (N = 71, 60.68%) that included the subjects with stationary or decreased diameters.The increase group had lower initial transverse, longitudinal, and largest diameter versus the non-increase group: 0.45 ± 0.12 versus 0.57 ± 0.17 (*p* < 0.0001), 0.36 ± 0.11 versus 0.43 ± 0.13 (*p* = 0.004), and 0.46 ± 0.12 versus 0.6 ± 0.16 (*p* < 0.0001).The increase group versus the non-increase group had a larger period of surveillance: a median of 48 (24, 84) versus 32.5 (12, 72) months (*p* = 0.045) and presented a similar age, pituitary hormone profile, and tumour lateralisation profile at baseline.We found a median diameter change of +0.14 cm versus −0.03 cm in the increase versus the non-increase groups (*p* < 0.0001).A rather high percent of patients might experience PI diameter increase during a longer period of follow-up, including those with a smaller initial size, while the age at diagnosis does not predict the tumour growth. This might help practitioners for long-standing surveillance according to our mentioned duration of follow-up.

## Figures and Tables

**Figure 1 diseases-12-00240-f001:**
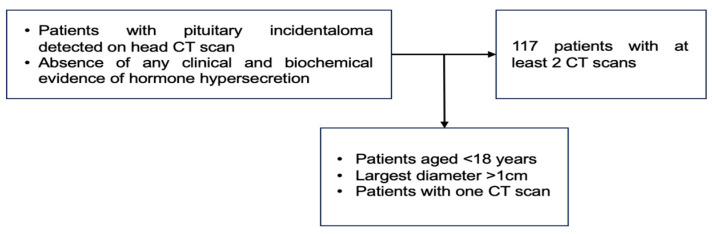
Study protocol. After initial confirmation of a PI, the patients underwent a second CT scan and an endocrine evaluation as well.

**Figure 2 diseases-12-00240-f002:**
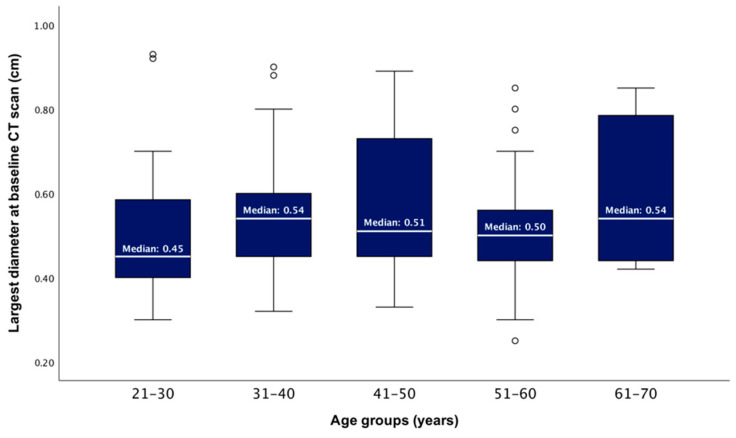
Boxplots showing the distributions of the baseline largest tumour diameter according to patients’ age groups (*p* = 0.334).

**Figure 3 diseases-12-00240-f003:**
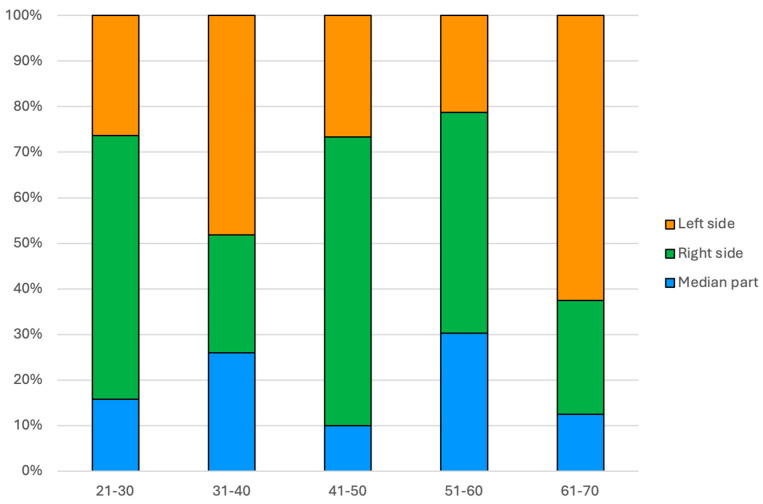
Bar chart showing the frequency of PI location within patients’ age groups (N = 117).

**Figure 4 diseases-12-00240-f004:**
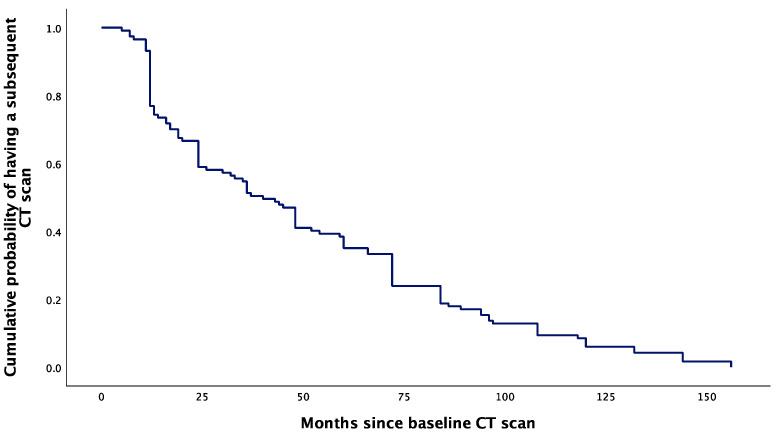
Kaplan–Meier curve showing the cumulative probability of having a subsequent CT scan after baseline evaluation for PI.

**Figure 5 diseases-12-00240-f005:**
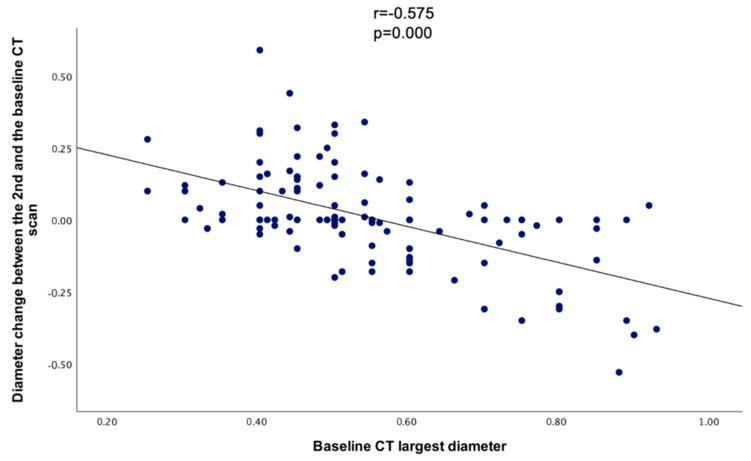
Scatterplot showing the correlation between the largest diameter at the baseline CT and the diameter change between the baseline and the 2nd CT.

**Figure 6 diseases-12-00240-f006:**
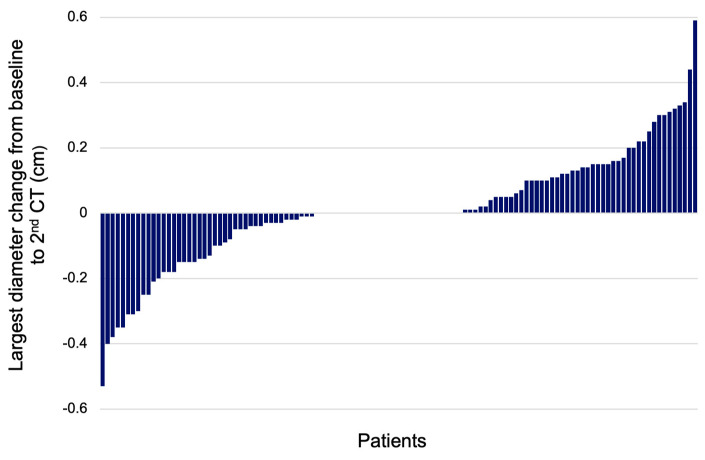
Waterfall plot showing the change of largest PI diameter between the baseline and 2nd CT scan (N = 117).

**Figure 7 diseases-12-00240-f007:**
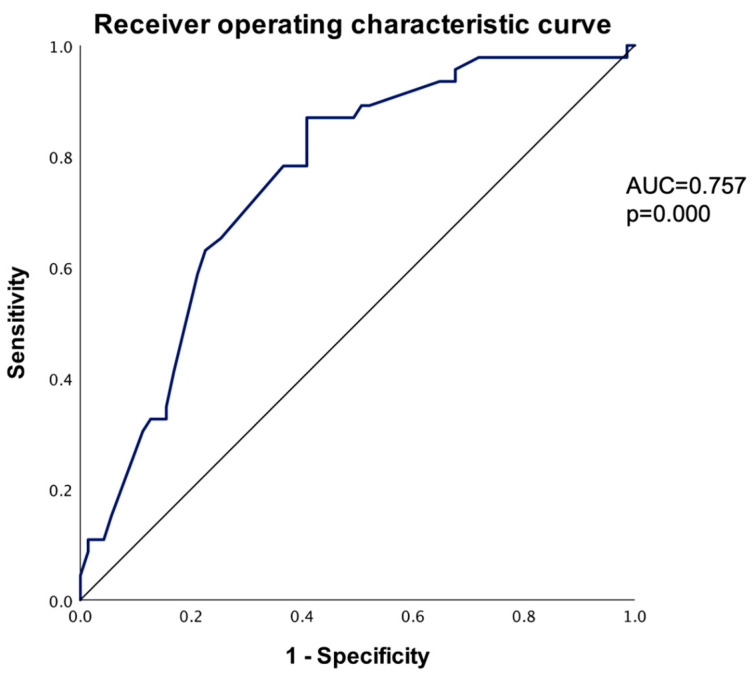
Receiver operating characteristic curve to predict PI increase based on largest diameter (smaller values of the test result variable indicate stronger evidence for a positive actual state).

**Table 1 diseases-12-00240-t001:** Baseline features of included adult population diagnosed with a non-functional PI less than 1 cm largest diameter.

Parameter	Value
**Age (years) mean ± SD**	43.86 ± 11.99
**Sex**	
Female, number of patients (%)	110 (94.02%)
Male, number of patients (%)	7 (5.98%)
**Baseline CT scan diameters**	
Transverse diameter (cm), mean ± SD	0.53 ± 0.16
Longitudinal diameter (cm), mean ± SD	0.41 ± 0.13
Largest diameter (cm), mean ± SD	0.55 ± 0.16
**Tumour location**	
Left side, number of patients (%)	38 (32.48%)
Right side, number of patients (%)	55 (47.01%)
Median part, number of patients (%)	24 (20.51%)
**Endocrine panel**	
FSH (mIU/mL), median (Q1, Q3)	33.40 (4.99, 67.24)
LH (mIU/mL), median (Q1, Q3)	5.66 (3.05, 44.18)
ACTH (pg/mL), mean ± SD	20.57 ± 13.09
Morning plasma cortisol (µg/dL), mean ± SD	12.79 ± 3.87
GH baseline (ng/mL), median (Q1, Q3)	0.30 (0.10, 0.76)
IGF1 baseline (ng/mL), mean ± SD	167.28 ± 42.31
Prolactin (ng/mL), mean ± SD	8.24 ± 4.46
TSH (µIU/mL), median (Q1, Q3)	1.40 (0.97, 2.34)

Abbreviations: ACTH = adrenocorticotropic hormone; FSH = follicle-stimulating hormone; GH = growth hormone; IGF1 = insulin-like growth factor 1; LH = luteinising hormone; CT = computed tomography; SD = standard deviation; TSH = thyroid stimulating hormone).

**Table 2 diseases-12-00240-t002:** Analysis based on the patients’ age groups (N = 117).

Age Group (Years)	N (% from the Entire Group)	Transverse Diameter Median (Q1, Q3)	Longitudinal Diameter Median (Q1, Q3)	Largest Diameter Median (Q1, Q3)	Left Side N (% from the Age Group)	Right Side N (% from the Age Group)	Median Part N (% from the Age Group)
21–30	9 (16.23)	0.45 (0.40, 0.59)	0.36 (0.31, 0.45)	0.45 (0.40, 0.59)	5 (26.32)	11 (57.89)	3 (15.79)
31–40	27 (23.07)	0.50 (0.40, 0.55)	0.40 (0.35, 0.50)	0.54 (0.45, 0.60)	13 (48.15)	7 (25.93)	7 (25.93)
41–50	30 (25.64)	0.51 (0.45, 0.73)	0.39 (0.32, 0.41)	0.51 (0.45, 0.73)	8 (26.67)	19 (63.33)	3 (10.00)
51–60	33 (28.20)	0.49 (0.49, 0.55)	0.35 (0.30, 0.48)	0.50 (0.44, 0.56)	7 (21.21)	16 (48.48)	10 (30.30)
61–70	8 (6.83)	0.54 (0.44, 0.79)	0.44 (0.35, 0.50)	0.54 (0.44, 0.79)	5 (62.50)	2 (25.00)	1 (12.50)

**Table 3 diseases-12-00240-t003:** CT-based PIs analysis between the baseline and second CT scan (N = 117).

Variable	Baseline CT Scan	2nd CT Scan	*p*-Value
**Increase group (N = 46, 39.32%)**			
Transverse diameter (cm), mean ± SD	0.45 ± 0.12	0.61 ± 0.16	**0.000**
Transverse diameter (cm), median (Q1, Q3)	0.45(0.40, 0.50)	0.60(0.52, 0.70)	
Longitudinal diameter (cm), mean ± SD	0.36 ± 0.11	0.42 ± 0.13	**0.003**
Longitudinal diameter (cm), median (Q1, Q3)	0.35(0.30, 0.40)	0.40(0.30, 0.53)	
Largest diameter (cm), mean ± SD	0.46 ± 0.12	0.62 ± 0.15	**0.000**
Largest diameter (cm), median (Q1, Q3)	0.45 (0.40, 0.50)	0.60 (0.53, 0.70)	
**Non-increase group (N = 71, 60.68%)**			
Transverse diameter (cm), mean ± SD	0.57 ± 0.16	0.49 ± 0.14	**0.000**
Transverse diameter (cm), median (Q1, Q3)	0.55 (0.45, 0.70)	0.49(0.40, 0.55)	
Longitudinal diameter (cm), mean ± SD	0.43 ± 0.13	0.39 ± 0.10	**0.001**
Longitudinal diameter (cm), median (Q1, Q3)	0.40 (0.33,0.50)	0.38(0.30, 0.45)	
Largest diameter (cm), mean ± SD	0.60 ± 0.16	0.51 ± 0.14	**0.000**
Largest diameter (cm), median (Q1, Q3)	0.56 (0.49, 0.75)	0.49 (0.40, 0.55)	
**Entire group (N = 117, 100%)**			
Transverse diameter (cm), mean ± SD	0.52 ± 0.01	0.54 ± 0.01	0.253
Transverse diameter (cm), median (Q1, Q3)	0.50 (0.42, 0.60)	0.53 (0.42, 0.63)	
Longitudinal diameter (cm), mean ± SD	0.40 ± 0.01	0.40 ± 0.01	0.727
Longitudinal diameter (cm), median (Q1, Q3)	0.38 (0.32, 0.46)	0.39 (0.30, 0.47)	
Largest diameter (cm), mean ± SD	0.54 ± 0.14	0.55 ± 0.01	0.453
Largest diameter (cm), median (Q1, Q3)	0.50 (0.44, 0.60)	0.54 (0.45, 0.66)	

**Table 4 diseases-12-00240-t004:** Analysis of the increase group (N = 59) versus the non-increase group (N = 61) at baseline and during follow-up.

Variable	Increase Group	Non-Increase Group	*p*-Value
**Number (%)**	46 (39.32%)	71 (60.68%)	
**Age (years), mean ± SD**	43.57 ± 12.22	44.06 ± 11.92	0.83
**Baseline CT scan diameters**			
Transverse diameter (cm), mean ± SD	0.45 ± 0.12	0.57 ± 0.17	**0.000**
Transverse diameter (cm), median (Q1, Q3)	0.45 (0.40, 0.50)	0.55 (0.45, 0.70)	
Longitudinal diameter (cm), mean ± SD	0.36 ± 0.11	0.43 ± 0.13	**0.004**
Longitudinal diameter (cm), median (Q1, Q3)	0.35 (0.30, 0.41)	0.40 (0.33, 0.50)	
Largest diameter (cm), mean ± SD	0.46 ± 0.12	0.60 ± 0.16	**0**
Largest diameter (cm), median (Q1, Q3)	0.45 (0.40, 0.50)	0.56 (0.49, 0.75	
**2nd CT scan diameters**			
Transverse diameter (cm), mean ± SD	0.61 ± 0.16	0.50 ± 0.14	**0**
Transverse diameter (cm), median (Q1, Q3)	0.60 (0.52, 0.70)	0.49 (0.40, 0.55)	
Longitudinal diameter (cm), mean ± SD	0.42 ± 0.13	0.39 ± 0.10	**0**
Longitudinal diameter (cm), median (Q1, Q3)	0.40 (0.30, 0.53)	0.38 (0.30, 0.45)	
Largest diameter (cm), mean ± SD	0.62 ± 0.15	0.51 ± 0.14	**0**
Largest diameter (cm), median (Q1, Q3)	0.60 (0.55, 0.70)	0.49 (0.41, 0.55)	
**Largest diameter change between baseline 2nd CT scan**			
Largest diameter change (cm), mean ± SD	0.17 ± 0.13	−0.09 ± 0.12	**0**
Largest diameter change (cm), median (Q1, Q3)	0.14 (0.07, 0.22)	−0.03 (−0.15, 0.00)	
**Tumour location**			
Left side, number (%)	19 (41.30%)	19 (26.76%)	0.088
Right side, number (%)	16 (34.78%)	39 (54.93%)	
Median part, number (%)	11 (23.91%)	13 (18.31%)	
**Endocrine panel**			
FSH (mIU/mL), median (Q1, Q3)	36.38 (6.99, 64.91)	23.75 (4.69, 79.41)	0.846
LH (mIU/mL), median (Q1, Q3)	13.48 (3.06, 52.48)	5.66 (2.02, 35.14)	0.418
ACTH (pg/mL), median (Q1, Q3)	16.98 (12.00, 25.64)	19.77 (12.04, 32.95)	0.537
Morning plasma cortisol (µg/dL), mean ± SD	11.64 ± 3.39	13.73 ± 4.08	0.138
GH baseline (ng/mL), mean (Q1, Q3)	0.51 (0.11, 0.90)	0.24 (0.06, 0.60)	0.261
IGF1 baseline (ng/mL), mean ± SD	149.75 (133.35, 205.80)	169.80 (147.15, 191.05)	0.862
Prolactin (ng/mL), mean ± SD	8.19 ± 4.59	8.30 ± 4.42	0.93
TSH (µIU/mL), median (Q1, Q3)	1.21 (0.90, 1.74)	1.94 (1.11, 3.17)	0.28
**Surveillance duration**			
Months between baseline 2nd CT, median (Q1, Q3)	48.00 (24.00, 84.00)	32.50 (12.00, 72.00)	**0.045**

Abbreviations: ACTH = adrenocorticotropic hormone; FSH = follicle-stimulating hormone; GH = growth hormone; IGF1 = insulin-like growth factor 1; LH = luteinising hormone; CT = computed tomography; SD = standard deviation; TSH = thyroid stimulating hormone.

**Table 5 diseases-12-00240-t005:** Time frame analysis between the two assessments (months).

Age Group	Months of Follow-Up	
	Mean ± SD	Median (Q1, Q3)
21–30 years	55.26 ± 10.11	43.00 (17.00, 89.00)
31–40 years	52.33 ± 8.12	36.00 (16.00, 84.00)
41–50 years	45.63 ± 6.85	36.00 (12.00, 67.50)
51–60 years	51.06 ± 7.02	48.00 (12.50, 78.00)
61–70 years	55.25 ± 13.23	60.00 (15.00, 81.00)
**Entire group**	**50.93 ± 39.97**	**40.00 (13.50, 72.00)**
**Increase group**	**58.11 ± 39.19**	**48.00 (24.00, 84.00)**
**Non-increase group**	**46.28 ± 40.05**	**32.50 12.00, 72.00)**

**Table 6 diseases-12-00240-t006:** Analysis based on using an arbitrary cut-off value of 0.5 cm.

Cut-Off Value	Sensitivity	Specificity	Youden Index
0.50 cm	65.21%	74.64%	0.398
0.545 cm	87.00%	59.20%	0.462

**Table 7 diseases-12-00240-t007:** Sensitivity and specificity for the 0.50 cm and 0.545 cm cut-off values.

Cut-Off Value	Sensitivity	Specificity	Youden Index
0.50 cm	65.21%	74.64%	0.398
0.545 cm	87.00%	59.20%	0.462

## Data Availability

The research data supporting this study’s findings are not publicly available. Further enquiries can be directed to the corresponding author.

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
