# Peer review of "A Real-World Longitudinal Study in Non-Functioning Pituitary Incidentalomas: A PRECES Micro-Adenomas Sub-Analysis"

_diseases, 2024, doi:10.3390/diseases12100240_

Round 1
Reviewer 1 Report
Comments and Suggestions for Authors
The authors have provided a retrospective analysis of CT evaluations of pituitary incidentalomas.
The authors should provide a hypothesis to assist clinicians in decision making. If all they are presenting is that adenomas may become bigger or smaller then there is no new information.
Much of the data appears as a regression to the mean, as shown in the scatter plot (which should be figure 5 rather than figure 4) and in the waterfall plot (which should be figure 6 rather than figure 5). What the investigators need to demonstrate is the least significant change in CT evaluation of the size of adenoma by preferably three separate blinded independent investigators.
In table 2 and figure 2 they must explain the rationale of transverse diameter versus longitudinal diameter versus largest diameter. For the clinician the largest diameter would define the size of the tumor. Also, they need to explain why they are using median and not mean, and to give reason that they are not using normal distributions.
In figure 3, is there any reason for commenting on left side, right side or median? Are the relative distributions according to age significant by ANOVA or is this just a random graph? If it is a random distribution and there is no clinical value, the data in the results session section should be eliminated.
Figure 4 is the observation of obtaining a repeat CT scan. Unless there is some reason to suggest that the rate of change in the size of the nodule is related to the months of the CT scan this figure in the data in the results section should be eliminated.
Figure 6 which is mislabeled figure 5 the waterfall plot looks like regression to the mean. That is the same number of nodules are getting bigger as are getting smaller. This should be explained in the results section.
Figure 5 which is mislabeled as Figure 4 appears that if you draw a line at 0 that baseline larger diameter lesions get smaller and the smaller diameter lesions get bigger again this suggests a regression to the mean.
Table 3 is meaningless. If you define your groups as increased or decreased then they will be statistically different because of the definition.
There does not seem to be any clinical indication for the data in Table 5
Both tables 6 and 7 are mislabeled and should read “sensitivity.” It is not clear what the receiver operating curves are trying to demonstrate sensitivity and specificity for what. This should be explained in the results and discussion.
In summary, the authors need to present a hypothesis and present which clinical parameters which may help clinicians decide upon repeating the CT scans, or they may conclude that incidentalomas followed for a specified time remain microadenomas. The latter would be a more focused and much shorter presentation.
Comments on the Quality of English LanguageGrammatical errors.
Author Response
Response to Review 1 Comments
Dear Reviewer,
Thank you very much for your time and your effort to review our manuscript.
We are very grateful for providing your valuable feedback on the article.
Here is our response and related amendment that has been made in the manuscript according to your review (marked in yellow color).
The authors have provided a retrospective analysis of CT evaluations of pituitary incidentalomas.
Thank you very much. We really appreciate it!
The authors should provide a hypothesis to assist clinicians in decision making. If all they are presenting is that adenomas may become bigger or smaller then there is no new information.
Thank you very much. According to your recommendation we added it: “Our hypothesis was that during more than 3 years of mean follow-up duration of imagistic surveillance, some patients might experience micro-adenoma increase including in smaller tumors, but no significant clinical impact is expected (in terms of remaining non-functioning, not becoming neurosurgery candidates due to massive PI increase and becoming larger than 1 cm or associating a pituitary apoplexy).” Also, we confirmed the hypothesis based on the results section. Thank you
Much of the data appears as a regression to the mean, as shown in the scatter plot (which should be figure 5 rather than figure 4) and in the waterfall plot (which should be figure 6 rather than figure 5). What the investigators need to demonstrate is the least significant change in CT evaluation of the size of adenoma by preferably three separate blinded independent investigators.
Thank you very much. According to your recommendation, we corrected these figures numbers.
With regard to the regression to the mean, this is not unusual in real-life studies. However, in this study, we tested the linear regression between the largest CT diameter at baseline and the diameter at the 2nd CT scan and the result showed a statistically significant correlation coefficient, of r=0.394, p<0.001, thus regression toward the mean does not seem a statistical phenomenon to impact our sample which is rather large. Thank you
In table 2 and figure 2 they must explain the rationale of transverse diameter versus longitudinal diameter versus largest diameter. For the clinician the largest diameter would define the size of the tumor. Also, they need to explain why they are using median and not mean, and to give reason that they are not using normal distributions.
Thank you very much. All these 3 diameters are provided and used in daily practice in the hospitals that provided the data amid this study. As there were small sub-groups, the tests of normality had a significance value below 0.05 and therefore these distributions were considered non-Gaussian. Therefore, the central tendencies were best described by quartiles/median, not by mean, as explained in Methods section. Thank you
In figure 3, is there any reason for commenting on left side, right side or median? Are the relative distributions according to age significant by ANOVA or is this just a random graph? If it is a random distribution and there is no clinical value, the data in the results session section should be eliminated.
Thank you very much. We analyzed the associations between the categorical variables, the appropriate test being Fisher’s exact test. We generated the p-value and added the detailed explanation into the results section. “Regarding the PIs side amid patients’ age groups, in the 31-40 years age group, there were significantly more PIs located on the left side than expected (+2.0 adjusted residual), and fewer on the right side (-2.5 adjusted residual). In the 41-50 years age group, there were more PIs located in the right part than expected (+2.1 adjusted residual, p=0.045). No ANOVA test was used since the chart does not introduce numeric variables, only sub-groups. Thank you
Figure 4 is the observation of obtaining a repeat CT scan. Unless there is some reason to suggest that the rate of change in the size of the nodule is related to the months of the CT scan this figure in the data in the results section should be eliminated.
Thank you very much. In this figure, we aimed to show that the patients did not present to follow-up at specific time intervals according to a certain protocol, as the study was done in the real-life setting and clinicians had to adjust the assessment to the patients’ circumstances (for instance, re-scan for a prior head trauma, etc.). We found no correlation between the rate of PI change and the months between the scans (r=0.074 si p=0.104). Thank you
Figure 6 which is mislabeled figure 5 the waterfall plot looks like regression to the mean. That is the same number of nodules are getting bigger as are getting smaller. This should be explained in the results section.
Thank you very much. We respectfully mention that we changed the figure number accordingly, while referring to the regression to the mean, the x axis does not sort the values depending on the baseline largest CT diameter, but on the change between the baseline and the 2nd CT scan, and thus we respectfully consider that the figure does not suggest the fact that the extreme values are closer to the mean on the second measurement as pinpointed by the mentioned phenomenon. Thank you very much.
Figure 5 which is mislabeled as Figure 4 appears that if you draw a line at 0 that baseline larger diameter lesions get smaller and the smaller diameter lesions get bigger again this suggests a regression to the mean.
Thank you very much. We changed the figure number accordingly. As answered above. Thank you
Table 3 is meaningless. If you define your groups as increased or decreased, then they will be statistically different because of the definition.
Thank you very much. We introduced Table 3 to pinpoint the entire group as well. Thank you
There does not seem to be any clinical indication for the data in Table 5
Thank you very much. We specified at limits of the study that the circumstances of imaging scanning were not analyzed in this study. Thank you
Both tables 6 and 7 are mislabeled and should read “sensitivity.” It is not clear what the receiver operating curves are trying to demonstrate sensitivity and specificity for what. This should be explained in the results and discussion.
Thank you very much. We changed “sensibility” with “sensitivity”. “As noted in Table 4, the baseline largest CT diameter may be the strongest predictor for a PI growth, therefore, using the baseline CT diameter we plotted the ROC curve: for each possible cut-off point, we generated the sensitivity and 1-specificity chart with a significant area under the curve of 0.757, demonstrating the strength of the baseline largest CT diameter as a predictor. The ROC curve shows the sensitivity and specificity for the PI increase. (Figure 7)” Thank you
In summary, the authors need to present a hypothesis and present which clinical parameters which may help clinicians decide upon repeating the CT scans, or they may conclude that incidentalomas followed for a specified time remain microadenomas. The latter would be a more focused and much shorter presentation.
Thank you very much. We answered above and adjusted the Conclusions. Thank you
Comments on the Quality of English Language: Grammatical errors.
Thank you very much. We corrected it.
Thank you very much
Reviewer 2 Report
Comments and Suggestions for Authors
The article titled "A real-world longitudinal study in non-functioning pituitary incidentalomas (a PRECES sub-analysis)" presents a multi-centric, longitudinal, and retrospective study that investigates dynamic changes in non-functioning pituitary incidentalomas (PIs) in adults. I regret to inform you that the study has several limitations:
The study specifically includes patients with incidentally discovered tumors smaller than 1 cm. However, the title does not clearly indicate this detail, and a more descriptive title would better reflect the study's focus and the specific patient population, providing immediate clarity to readers.
The study's reliance on CT scans for the follow-up of pituitary incidentalomas is a significant limitation. CT is not the recommended imaging modality for these tumors due to its relatively low specificity and the risks associated with radiation exposure. The tumor diameters measured in the study are also questionable, particularly given the inclusion of tumors smaller than 1 cm. Accurate measurement of such small tumors is challenging, especially when using CT, and this raises concerns about the reliability of the data.
Additionally, the study cohort is overwhelmingly female (94.02%), which greatly limits the generalizability of the findings, particularly to male patients.
Another concern is the lack of a standardized follow-up protocol across participating centers, leading to variability and potential bias in the results.
Furthermore, the study does not offer actionable guidelines or recommendations based on its findings.
Author Response
Response to Review 2 Comments
Dear Reviewer,
Thank you very much for your time and your effort to review our manuscript.
We are very grateful for your insightful comments and observations, also, for providing your valuable feedback on the article.
Here is a point-by-point response and related amendments that have been made in the manuscript according to your review (marked in yellow color).
The article titled "A real-world longitudinal study in non-functioning pituitary incidentalomas (a PRECES sub-analysis)" presents a multi-centric, longitudinal, and retrospective study that investigates dynamic changes in non-functioning pituitary incidentalomas (PIs) in adults.
Thank you very much.
I regret to inform you that the study has several limitations:
Thank you very much. We addressed your observations and recommendations as follows:
The study specifically includes patients with incidentally discovered tumors smaller than 1 cm. However, the title does not clearly indicate this detail, and a more descriptive title would better reflect the study's focus and the specific patient population, providing immediate clarity to readers.
Thank you very much. We followed your recommendation and corrected the title: “A real-world longitudinal study in non-functioning pituitary incidentalomas: (a PRECES micro-adenomas sub-analysis)”. Thank you
The study's reliance on CT scans for the follow-up of pituitary incidentalomas is a significant limitation. CT is not the recommended imaging modality for these tumors due to its relatively low specificity and the risks associated with radiation exposure.
Thank you very much. With respect to this useful point, we mention the following aspects:
MRI is preferred to CT which is the second option, but, in many centers (as seen in our country) CT scan is more often available and it has a free reimbursement for certain medical and surgical circumstances/conditions, thus the study may be useful for similar centers as ours.
Some patients may have contraindications to MRI due to, for instance, bone synthesis materials as used in fractures that have been surgically treated; thus, only CT may be used in these circumstances. Moreover, we mentioned as limits: “Our retrospective data collection did not allow observing all the circumstances for which the CT scan was initially done (such as trauma, infections, headache, etc.).”
In addition, CT scan may have been done for unrelated conditions to the pituitary tumors that might be easily explored by CT exam such as headache, trauma, control for prior infections (that also allowed the evaluation of the pituitary incidentalomas, including during follow-up of those ailments).
Overall, as mentioned, the limits of having CT rather than MRI examination are have been specified, but this type of investigation amid real-life setting and not routine endocrine protocols is used in many hospitals, particularly in multidisciplinary centers whereas the patients, as expected for incidentalomas scenario, have other unrelated ailments suitable for a CT exploration.
Thank you
The tumor diameters measured in the study are also questionable, particularly given the
inclusion of tumors smaller than 1 cm. Accurate measurement of such small tumors is challenging, especially when using CT, and this raises concerns about the reliability of the data.
Thank you. We mentioned the data were analyzed by the imaging team in each hospital and then confirmed by the imaging analysis amid the study:
“A second check-up CT analysis was further processed (by dr. MC) after imaging data was registered in each center in order to confirm the PIs size and achieve a homogenous interpretation of all the data included this study.”
Thank you
Additionally, the study cohort is overwhelmingly female (94.02%), which greatly limits the generalizability of the findings, particularly to male patients.
Thank you very much. Indeed, these are the results we obtained amid real-world settings. We followed your recommendation and added this aspect as limits of the real-life study: “The study cohort was overwhelmingly female (94.02%), which might limit the generalizability of the findings to male patients.”
Thank you
Another concern is the lack of a standardized follow-up protocol across participating centers, leading to variability and potential bias in the results.
Thank you very much. We mentioned this aspect which represents a particular perspective of this real-world setting that is commonly found nowadays. For example, one patient who had a head trauma and underwent a CT scan might have been diagnosed with a pituitary micro-incidentaloma and the same had the indication of re-exam after 6 months (check-up) and under these circumstances, the pituitary tumor was re-checked as well.
We respectfully introduced this point at limits of the study in Discussion section: “As limits of the study we mention the real-world data collection that may bring a level of bias since no specific surveillance protocol was used in each subject.”
Thank you
Furthermore, the study does not offer actionable guidelines or recommendations based on its findings.
Thank you very much. This was a retrospective study and the conclusions have been listed at the end of the article. This work was not intended to be a guideline by its design.
For instance:
“This was a longitudinal study in 117 adults (aged between 20 and 70 years) diagnosed with non-functioning micro-PIs followed for a mean period of 50.93 months.”
“No pituitary incidentaloma became functioning during follow-up, neither associated hypopituitarism or increased beyond the diameter of 1 cm; no case of pituitary apoplexy was found.”
“The analysis based on patients’ decades of age showed that most of them were between the age of 31 and 60 years with similar diameters.”
“39.32% of the patients had a larger diameter during follow-up (increase group) versus non-increase group (60.68%) that included the subjects with stationary or decreased diameters.”
“A rather high percent of patients might experience diameter increase during a longer period of follow-up, including those with a smaller initial size, while the age at diagnosis does not predict the tumour growth. This might help practitioners for long standing surveillance according to our mentioned duration of follow-up.”
Thank you
Thank you very much
Round 2
Reviewer 1 Report
Comments and Suggestions for Authors
In conclusion spell out pituitary incidentaloma.
Comments on the Quality of English Languageneeds editorial review.